REGISTERED REPORT PROTOCOL

# Implementation of an electronic patient-reported measure of barriers to antiretroviral therapy adherence with the Opal patient portal: Protocol for a mixed method type 3 hybrid pilot study at a large Montreal HIV clinic

**Kim Engler** [1] *, **Serge Vicente** [2], **Yuanchao Ma** [1], **Tarek Hijal** [3], **Joseph Cox** [4,5], **Sara Ahmed** [6], **Marina Klein** [1,4,7], **Sofiane Achiche** [8], **Nitika Pant Pai** [7], **Alexandra de Pokomandy** [1,4,9], **Karine Lacombe** [10], **Bertrand Lebouché** [1,4,9]

1 Center for Outcomes Research and Evaluation, Research Institute of the McGill University Health Centre, Montreal, Quebec, Canada, 2 Department of Mathematics and Statistics, University of Montreal, Montreal, Quebec, Canada, 3 Department of Radiation Oncology, Cedars Cancer Center, McGill University Health Centre, Montreal, Quebec, Canada, 4 Division of Infectious Disease, Department of Medicine, Chronic Viral Illness Service, McGill University Health Centre, Montreal, Quebec, Canada, 5 Department of Epidemiology, Biostatistics, and Occupational Health, McGill University, Montreal, Quebec, Canada, 6 School of Physical & Occupational Therapy, McGill University, Montreal, Quebec, Canada, 7 Department of Medicine, McGill University, Montreal, Quebec, Canada, 8 Department of Mechanical Engineering, École Polytechnique de Montréal, Montreal, Quebec, Canada, 9 Department of Family Medicine, McGill University, Montreal, Quebec, Canada, 10 Sorbonne Université, Inserm IPLESP, Hôpital St Antoine, Assistance Publique -Hôpitaux de Paris, Paris, France

* kimcengler@gmail.com

This is a Registered Report and may have an associated publication; please check the article page on the journal site for any related articles.

## Abstract

### Background

Adherence to antiretroviral therapy (ART) remains problematic. Regular monitoring of its barriers is clinically recommended, however, patient-provider communication around adherence is often inadequate. Our team thus decided to develop a new electronically administered patient-reported outcome measure (PROM) of barriers to ART adherence (the I-Score) to systematically capture this data for physician consideration in routine HIV care. To prepare for a controlled definitive trial to test the I-Score intervention, a pilot study was designed. Its primary objectives are to evaluate patient and physician perceptions of the I-Score intervention and its implementation strategy.

### Methods

This one-arm, 6-month study will adopt a mixed method type 3 implementation-effectiveness hybrid design and be conducted at the Chronic Viral Illness Service of the McGill University Health Centre (Montreal, Canada). Four HIV physicians and 32 of their HIV patients with known or suspected adherence problems will participate. The intervention will involve

**Data Availability Statement:** All relevant quantitative data from this study will be made available upon study completion.

**Funding:** BL, KE, SA, and MK received pilot funding from the CIHR Canadian HIV Trials Network (https://www.hivnet.ubc.ca/) to conduct this study (Grant # CTNPT039). BL and KE received funding from Merck Canada Inc. to develop the I-Score PROM under their Investigator-Initiated Study Program (Grant # IISP-53538). BL received funding from the Quebec Strategy for Patient-Oriented Research (SPOR) Support Unit (Methodological Developments) also to develop the I-Score PROM (Grant # M006). BL received funding from Merck Canada Inc./MSD France to configure the Opal patient portal for HIV care at the Chronic Viral Illness Service of the McGill University Health Centre (Grant # 65364). KL and BL received funding from MSD Avenir to lead a Parisian trial modeled on this pilot study (Grant # DS-2018-0072). The funders had and will not have a decisional role in study design, data collection, data analysis, or preparation of manuscripts for publication based on its results.

**Competing interests:** BL has received research support and consulting fees from ViiV Healthcare, Merck, and Gilead. This does not alter our adherence to PLOS ONE policies on sharing data and materials.

having patients complete the I-Score through a smartphone application (Opal), before meeting with their physician. Both patients and physicians will have access to the I-Score results, for consideration during the clinic visits at Times 1, 2 (3 months), and 3 (6 months). The implementation strategy will focus on stakeholder involvement, education, and training; promoting the intervention's adaptability; and hiring an Application Manager to facilitate implementation. Implementation, patient, and service outcomes will be collected (Times 1-2-3). The primary outcome is the intervention's acceptability to patients and physicians. Qualitative data obtained, in part, through physician focus groups (Times 2–3) and patient interviews (Times 2–3) will help evaluate the implementation strategy and inform any methodological adaptations.

## Discussion

This study will help plan a definitive trial to test the efficacy of the I-Score intervention. It will generate needed data on electronic PROM interventions in routine HIV care that will help improve understanding of conditions for their successful implementation.

## Clinical trial registration

ClinicalTrials.gov identifier: NCT04702412; https://clinicaltrials.gov/.

## Introduction

Routinely collecting data on patient-reported outcome measures (PROMs) for individual patient care can benefit both people living with HIV and their providers, yet it is seldom done in HIV clinical practice [1]. For patients, it may help ensure that HIV care is person-centered and in line with their needs [1]. For providers, given the multidimensional and chronic nature of HIV clinical assessment and follow-up, the use of PROMs could facilitate efficient application of clinical guidelines in a context of time and resource constraints [2].

While past syntheses of effectiveness evidence for PROM use across specialties in routine care have typically found mixed results, with inconsistent impacts on patient outcomes [3,4], a more recent systematic review published in 2019 finds the evidence supports PROM use in standard care, particularly to improve patient-provider communication and decision-making in clinical practice [5]. Furthermore, the international momentum building for PROM use [6] may increase with the current COVID-19 pandemic. Indeed, there are calls for a scale up of electronic PROM implementation in this crisis for the remote follow-up of chronic conditions, in part, to better screen patients and promptly manage their needs [7].

The management of antiretroviral therapy (ART) adherence for the treatment of HIV is among the areas that could profit from greater PROM use. Successful ART remains essential to a near-normal life expectancy; however, many on ART have suboptimal adherence [8,9], even on single-tablet regimens [10,11]. In a recent study, only 23% percent of adults initiating a single-tablet regimen were considered adherent over a six-month period versus 12% among those who initiated a multiple tablet regimen, based on prescription fill dates [10]. Clinically recommended strategies to foster adherence include ongoing monitoring of barriers to adherence among people living with HIV [12]. Yet, several studies point to inadequate patient-provider communication around ART adherence and its impediments [13–17] and many HIV providers underestimate their patients' adherence difficulties [18,19]. In addition, individuals

with HIV collectively report a multitude of barriers to adherence, including a variety of cognitive, emotional, social, and material issues as well as health service-related barriers [20,21], the proper evaluation of which may be time-consuming for providers [22].

For these reasons, with patient [23] and provider [24] involvement, we are developing a PROM of barriers to ART adherence, the Interference-Score (or I-Score), for electronic administration. I-Score data will be collected from patients and shared with their providers via Opal, a patient portal and smartphone app. This award-winning app [25], which is currently in use at the Cedars Cancer Centre of the McGill University Health Centre (MUHC), will be configured to respond to the needs of patients with HIV. Opal can give patients access to appointment schedules, laboratory test results, educational material, waiting room management tools, and PROMs. Electronic administration of our PROM was crucial as it simplifies score integration within the clinical workflow, allows for longitudinal presentation of scores as well as remote monitoring, and through Opal, it provides access to several other useful and potentially empowering patient-centered functions.

## Aim and objectives

With the present mixed method pilot study, drawing on implementation science, the aim is to develop the methods and tools necessary to undertake a more robust evaluation of the implementation and effectiveness of the I-Score PROM-within-Opal innovation (henceforth, the I-Score intervention) in routine HIV care with individuals on ART. This study's primary objectives are to evaluate stakeholder perceptions of the I-Score innovation (Objective 1) and evaluate the implementation strategy (Objective 2) in terms of recommended implementation science metrics for PROMs in routine care [26]. Its secondary objective (Objective 3) is to determine if the intervention shows promise and the chosen outcomes are useful, by observing collected data on select effectiveness outcomes (patient and service outcomes).

## Guiding frameworks

It is important that a credible causal explanation of a digital health innovation's intended impacts be provided [27]. Indeed, in an electronic PROM-based intervention, conceptual or theoretical frameworks specify the mechanisms through which the intervention is expected to have its effects [28], facilitating appropriate outcome selection and the interpretation of results [29].

This pilot study will be guided, in part, by an intervention logic chain, depicted by the boxes in Fig 1, and adapted from the frameworks of Greenhalgh and colleagues [28,29]. The core of the intervention involves having patients complete the PROM prior to their HIV clinic visit and having both patients and providers receive and review the results. The left arrow in Fig 1 presents the key components of the implementation framework used, which will guide qualitative analysis. Specifically, these are the five broad domains of potential influence on implementation of Damschroder and colleagues' [30] Consolidated Framework for Implementation Research (CFIR) within which are grouped 39 distinct constructs. Hence, it is assumed that flow through the logic chain can be affected by features of the intervention, settings, individuals, and implementation process involved. The CFIR is a flexible and widely used framework in implementation research, including for PROM-based initiatives [26].

Another working framework (Fig 2) presents the broad hypothesized relationships between the implementation strategy used for the I-Score and the categories of study outcomes addressed. Borrowing from the frameworks of Stover and colleagues [26] and Santana and Feeny [31] it, in part, conceives successful implementation of I-Score use in standard HIV care, as potentially generating cascading effects on service and patient outcomes.

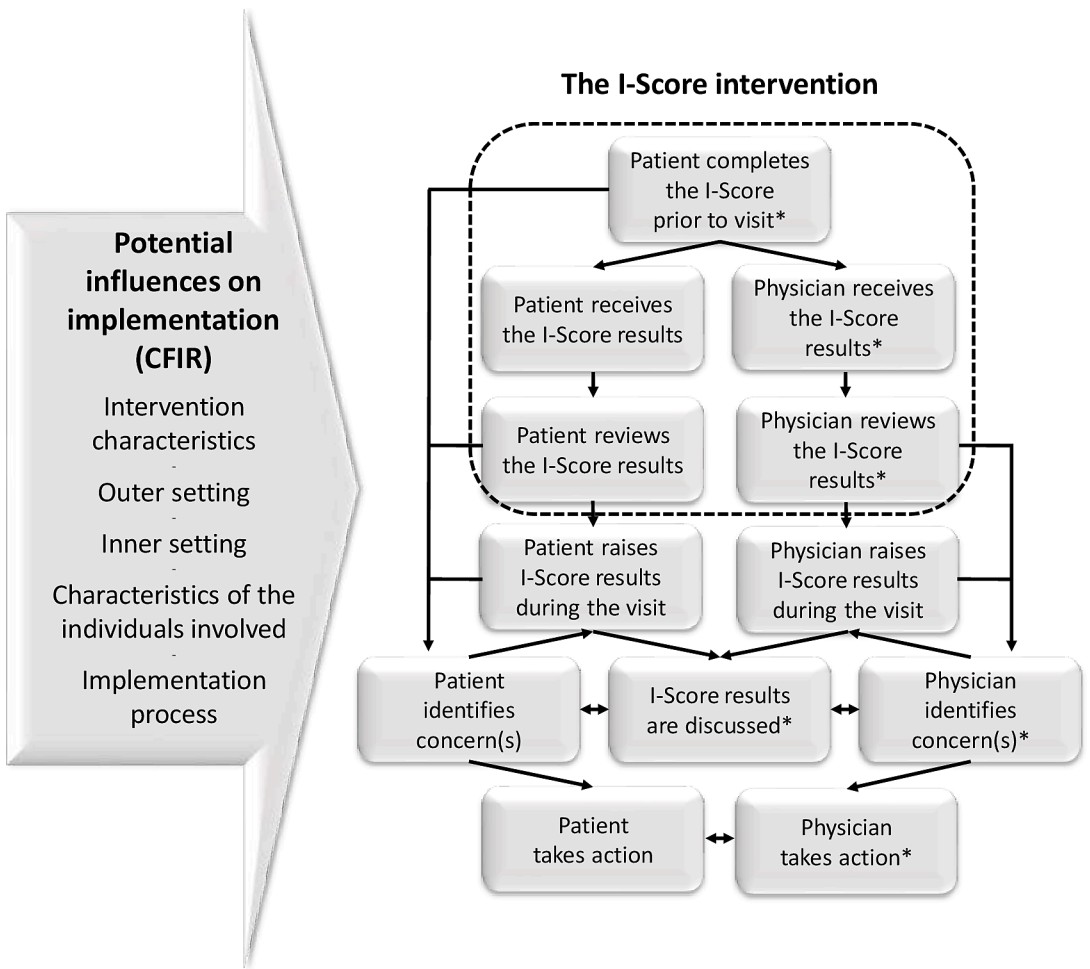

**Fig 1. Guiding implementation framework and I-Score intervention logic chain.** Asterisks indicate elements of the logic chain which will be examined as a part of this pilot study.

## Materials and methods

This study received approval by the McGill University Health Centre Research Ethics Board on January 18, 2021 (Study ID CTNPT039/ 2021–7190). Specifically, the Cells, Tissues, Genetics & Qualitative research panel approved the study.

### Study design

This 6-month pilot study will adopt a one-arm mixed method type 3 implementation-effectiveness hybrid design and be conducted in a single clinical site (Fig 3). Type 3 hybrid designs emphasize testing the implementation strategy of an evidence-based intervention, and to a lesser extent, reporting on intervention effectiveness [32].

Mixed methods were adopted in this study as multiple methods are recommended for studying intervention implementation and related challenges in complex systems, like HIV clinics [33]. The integration of the quantitative and qualitative data collected will occur toward study end within a convergent parallel design [34]. Those directly involved in the analyses will decide upon the specifics of integration. Reporting of this study will seek to satisfy the standards of Good Reporting of a Mixed Methods Study [35] and the Standards for Reporting Implementation Studies [36].

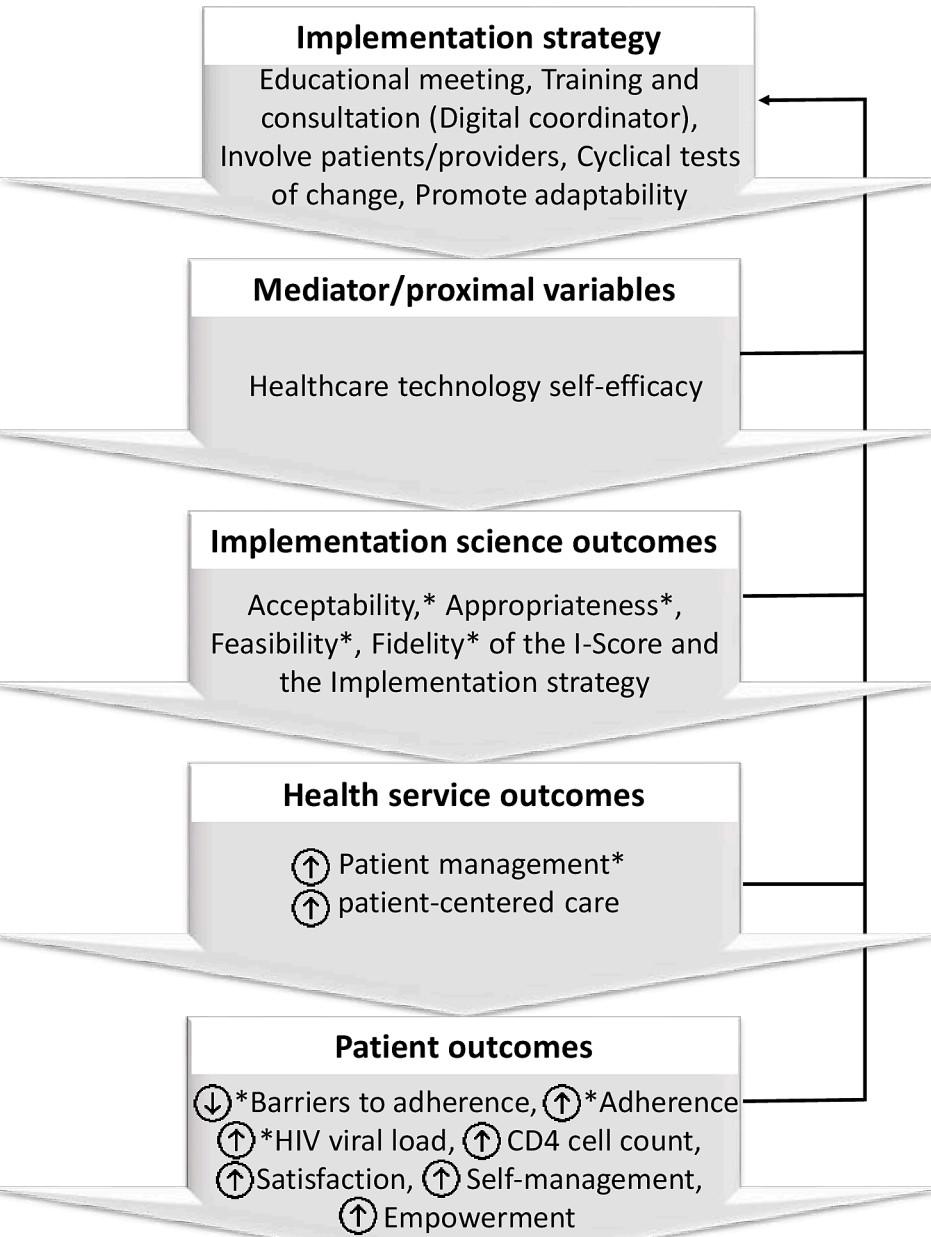

**Fig 2. Relationship between the I-Score implementation strategy and study outcomes.** Asterisks indicate outcomes for which data will be collected as a part of this pilot study.

## Setting and participants

The study setting is a large hospital-based clinic in Montreal, Quebec, Canada. This clinic, the Chronic Viral Illness Service (CVIS) of the MUHC, offers multidisciplinary care to over 1600 adults living with HIV. The CVIS and several team members have experience with implementation science methods and related pilot studies [e.g., 37].

Among the 16 physicians actively treating individuals with HIV at the CVIS, four will be recruited to participate as well as 32 of their adult patients. This sample size amply meets rule

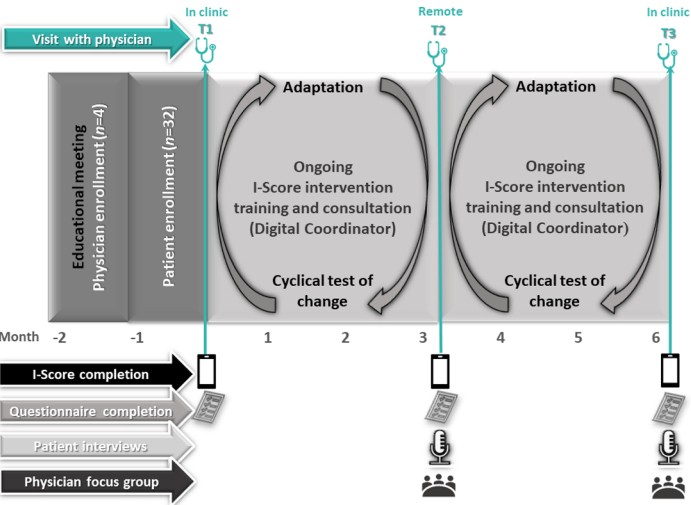

**Fig 3. Pilot study design.**

of thumb recommendations for one-arm pilot studies [38]. To participate, patients must be confirmed HIV positive, aged at least 18 years old, on combination ART, irrespective of duration, and literate in English or French. They must own a smartphone with an appropriate data plan and/or home Wi-Fi connection, since currently, Opal is ideally suited to a smartphone interface. They must also be willing to download the smartphone app. Finally, patients must have had known or suspected adherence problems in the past 12 months, based on: a detectable viral load test result per local standards [39], and/or report by the patient or healthcare team (by the physician, nurse, social worker, or pharmacist). At least ten female patients among the 32 will be recruited, to ensure sufficient representation of women living with HIV [40]. Patients may not participate if they are: concurrently enrolled in a clinical trial; affected by a cognitive impairment or medical instability that prevents them from participating in all aspects of the study; insufficiently able to use the app with the technical support provided; receiving treatment for hepatitis C or have completed treatment 3 months ago or less; or being treated for hepatitis B with a medication other than their combination ART.

**Recruitment and consent process.** Physicians and patients are expected to be enrolled from July to August 2021. Physicians treating patients with HIV at the CVIS will be asked individually to participate, by oral (e.g., phone) or email invitation. Patients of participating physicians will be recruited and consented in two ways: 1) when they visit the clinic; or 2) prior to an upcoming clinic visit. When they visit the clinic to meet with a health or social service provider (e.g., physician, nurse, social worker, psychologist), the provider will briefly describe the study to determine interest. Alternatively, suitable patients with upcoming visits will be identified by the physician and then contacted by a neutral clinic staff member to inform them of the study. If the individual is interested, the study coordinator will present the project in greater detail to them in person or over the phone, check the eligibility criteria, and obtain consent (See S1 Appendix for the patient consent form). Then, an appointment for a teleconference (e.g., on Zoom) or in person meeting at the clinic will be made with the participant prior to their next regular clinic appointment with their physician to deliver training to use the app and to complete the I-Score. The app's installation and functionality on the patient's smartphone will also be verified. Given efforts to limit in-hospital visits and risks for patients during the COVID-19 pandemic, in-clinic appointments with research staff will be avoided, where possible.

**The I-Score intervention.** During the study, participating patients will visit with their HIV physician three times, at Time 1 (T1), month 3 (T2), and month 6 (T3), prior to which they will complete the I-Score, as instructed. This visit schedule was selected as it concords with guidelines for the clinical follow-up of HIV in Quebec [41], while maximizing the collection of repeated study measurements. Given COVID-19, one visit (T2) will be done remotely (by phone or teleconferencing), while the other visits will be held at the CVIS. The I-Score PROM contains 20 items, covering 6 domains of barriers to ART adherence: cognitive and emotional aspects; lifestyle factors; the social and material context; the health experience and state; characteristics of ART; and the healthcare system and its services. Respondents indicate how often each barrier made adherence difficult in the past 4 weeks, with an 11-point scale, from 0% (never made it difficult) to 100% (always made it difficult). Details on the measure's development are published elsewhere [20,23,24,42]. Example items include "I was not motivated to take my medication," "I felt isolated or alone," "I had another health condition to deal with (for example, depression, diabetes, or heart disease)," and "My medication cost coverage was not sufficient."

The core intervention of this pilot consists of having individuals with HIV on ART register on the Opal app and complete the I-Score PROM prior to each of three consecutive visits with their HIV physician. The patients will receive a reminder to complete the I-Score one week before their visit and they will have immediate access to their results. The HIV physician will acquire the I-Score results before each visit via the ORMS dashboard, an appointment and questionnaire management tool integrated with Opal and designed specifically for healthcare providers. The option of graphically presenting scores over time will also be available, allowing the comparison of past and present scores. It is expected that patients and physicians will review the I-Score results so they can be considered during the clinic visit (Fig 1).

**Opal cybersecurity.** The technical cybersecurity aspects of the Opal app conform to the security and governance recommendations for the development of a patient portal, as identified by the MUHC's Security and Governance team, to ensure the confidentiality of patient data. For details, see the multimedia S2 Appendix of Kildea et al. [25].

**The implementation strategy.** The multilevel implementation strategy, designed for this study, addresses known facilitators and barriers to implementing electronic PROMs in routine clinical practice [43] and draws on recognized implementation strategies [44,45]. S1 Table presents the correspondence between the facilitators and barriers targeted, the chosen implementation strategies to address them, and their relationship to components of the implementation framework used in this study, the CFIR. As a part of our approach, we will conduct an "Educational meeting" by teleconference with providers to formally teach them about the intervention and its rationale and respond to concerns. We will provide "Training and consultation" on the PROM and app by hiring an "Application Manager" (AM). The AM will train patients and providers and be available to them on an ongoing basis, as needed, preferably by phone or teleconference. They will also help monitor the quality of PROM data. Appointing such a coordinator (or Quality Assurance officer) is a recommended strategy to minimize the impact of missing PROM data [46]. The AM will thus oversee the completeness of PROM data collected and manage any system or software problems, which are potential disadvantages of computer or web-based PROM administration [47]. Hence, overall, the AM will participate in the "Facilitation" of the PROM's implementation. Another strategy aims to meaningfully "Involve patients and providers" in the I-Score's implementation. Notably, following the I-Score administrations at T2 and T3, physicians will participate in a focus group (by teleconference), while a short semi-structured interview will be conducted with each patient (by telephone or teleconference). Throughout the study, the AM will take field notes on the problems encountered by participants and this feedback will enable "Cyclical small tests of change" to

improve implementation, using an evaluation approach guided by the Consolidated Framework for Implementation Research [48]. This way, we will "Promote adaptability" of the I-Score process, to enable adjustments to local considerations while maintaining the intervention's core components, namely, I-Score completion by the patient via Opal prior to the clinic visit and review of scores by the physician in conjunction with the visit. Many peripheral components will be adaptable, such as the timing and number of reminders to complete the I-Score and how I-Score results are presented to providers on the ORMS dashboard.

## Data collection

The data collection period is expected to extend from about September 2021 to February 2022.

**Quantitative component.** The quantitative component will have three sources of data: 1) participant self-report; 2) electronic medical records; and 3) passive data (e.g., on app use to assess fidelity). Patient self-report data will be collected via Opal. Physician self-report data will be obtained with paper questionnaires.

At T1, T2 (3 months) and T3 (6 months), a study questionnaire will be administered to participants, especially to assess implementation outcomes. At T1, the study questionnaire will be composed of 31 questions for patients and 24 for physicians, while at T2 and T3, it will have 21 questions for patients and 19 for physicians. Based on the metadata of patients who have used Opal, these questionnaires should take less than 10 minutes to complete, considering that patients take 10–15 seconds per question, at first completion. At T1, the questionnaire will ask about socio-demographics (e.g., year of birth, preferred language, sex, sexual orientation, ethnic group identity, immigration, education, income) and digital technology use as well as pose general health questions for patients (year of diagnosis with HIV, treatment satisfaction) and clinical practice questions for physicians (years practicing in HIV, current number of HIV patients) (For the full content of Time 1 study questionnaires, see S2 Appendix). The measures of digital technology use are as follows: frequency of mobile device use (adapted from Schnall et al. [49]), having a health app on one's mobile device [50], extent of health app use (adapted from Balapour et al. [51]), confidence in reporting medical information using mobile technology [51], and intention to report personal health data with a mobile device app, if asked by a provider [51]. This information will help contextualize the findings and describe the sample. Clinical data, namely, HIV viral load in copies/mL, to determine viral suppression, will be extracted from patients' medical health record at the clinic, at T1 and T3.

**Qualitative component.** The qualitative component will have three sources of data: 1) 1-hour focus groups with all physicians (T2, T3); 2) 45-minute interviews with patients (T2, T3), until core theme saturation (an intermediate sample of 15 should be sufficient at each time point [52]); and 3) the Application Manager field notes, recorded on a standardized form (T1-T3). Focus groups and patient interviews will be conducted by an experienced interviewer and, if possible, through a teleconferencing platform such as Zoom. Participants will have the option of accessing the teleconference by telephone or the Internet. The patient's name will not be shown. Audio recordings of the focus groups and interviews will be manually transcribed, extracting nominal information. Each will be guided by a similar semi-structured interview schedule, in English or French, depending on preferred language. It will ask about the participants' experience with I-Score use and its implementation as well as about facilitating and impeding factors. The schedule of study procedures for patients and physicians can be found in Table 1.

**Study metrics and instruments.** Details on the constructs assessed; the instruments and metrics used; the chosen thresholds for success, if applicable; the participant group contributing data; and the timing of data collection are presented in Table 2. Many of the chosen metrics are based on those recommended by Stover et al. [26]. Importantly, the authors emphasize the

**Table 1. Schedule of study procedures for participants.**

| Procedure | Timeline | | | |
|---|---|---|---|---|
| | Prior to study start | Study start (baseline) | Month 3 | Month 6 |
| Be screened and/or consented | Patients Physicians | | | |
| Attend educational meeting | Physicians | | | |
| Receive training on the I-Score measure and Opal | Patients Physicians | Patients Physicians | Patients Physicians | Patients Physicians |
| Complete the I-Score measure via Opal | | Patients | Patients | Patients |
| Examine the I-Score measure results via the ORMS dashboard | | Physicians | Physicians | Physicians |
| Attend HIV patient-physician visit (online or in person) | | Patients Physicians | Patients Physicians | Patients Physicians |
| Complete the post-visit checklist | | Physicians | Physicians | Physicians |
| Complete the online sociodemographic questionnaire | | Patients Physicians | | |
| Complete the online study questionnaire (after the clinic visit (s)) | | Patients Physicians | Patients Physicians | Patients Physicians |
| Possibly participate in an online qualitative interview (after the clinic visit) | | | Patients | Patients |
| Participate in an online focus group (after several clinic visits with participating patients) | | | Physicians | Physicians |
| Receive compensation | | Patients | Patients | Patients |

need to standardize evaluation metrics in patient-reported measure implementation and to distinguish between those used to assess perceptions of the innovation and those used to assess the implementation strategy. Not meeting the set thresholds for success, in this study, will signify that modifications are necessary before proceeding to a definitive trial [53].

**Objective 1 -evaluate perceptions of the I-Score intervention.** The primary outcome of this pilot study will be acceptability, as measured by the Acceptability E-scale (AES) for web-based PROMs (alpha coefficient: .76) [54]. Acceptability is related to how agreeable, palatable, or satisfactory an intervention is perceived to be by stakeholders [59]. It will be measured, at T1, T2, and T3, with an adapted version of the AES and administered to both patient and physician participants. The scale has 6 items rated on a 5-point Likert scale that varies depending on the item. Example items of the original measure include "How would you rate your overall satisfaction with this computer program?", and "How easy was this computer program [. . .] for you to use?" A summary score is obtained by adding the item scores (range: 6–30). A score of at least 24 (80% of maximum) indicates high acceptability and usability, as suggested by the scale developers.

Acceptability will also be measured at T1, T2 and T3 with a variant of the Net Promoters Score (NPS) used by England's National Health Service (NHS) and labelled the Friends and Family Test [55]. The NPS is considered a measure of user satisfaction [60]. A single question will be asked ("How likely are you to recommend the I-Score?") and rated on a 5-point Likert scale (1 = Extremely unlikely, 2 = Unlikely; 3 = Neither likely nor unlikely; 4 = Likely; 5 = Extremely likely). From this measure, the percentage recommending the I-Score will be calculated (score of 4 or 5), with a success threshold of 80% or more. An NPS-type score will also be calculated by creating three groups: promoters (score of 5), passives (score of 4), and detractors (score of 1–3). Subtracting the percentage of detractors from the promoters provides the NPS. NPS scores range from -100 to 100. A positive score ($> 0$) will be considered good [61], and a score of $\geq 50$, excellent.

Appropriateness concerns the perceived fit or relevance of the intervention for the particular users, setting, or problem at hand [59]. It will be measured, at T1, T2, and T3, with two

instruments. One concerns the perceived compatibility of the I-Score with the physicians' work. The perceived compatibility of an information technology innovation broadly relates to how consistent it is perceived to be with the potential users' values, needs, and past experiences [56]. It will only be collected from physicians, with a compatibility subscale developed by Moore and Benbasat (alpha coefficient: .86) [56]. It contains four items (e.g., "Using [the IT innovation] is compatible with all aspects of my work", "Using [the IT innovation] fits into my work style"). These are rated on a 7-point Likert scale, from Extremely disagree to Extremely agree, and averaged to produce the subscale score (range: 1–7). A minimum average score of 5.5 is the threshold set for compatibility.

In addition, a 4-item scale, the Appropriateness of Intervention Measure [57], will be completed by all participants (alpha coefficient: .91). Example items include "This [evidence-based practice] seems fitting" and "This [evidence-based practice] seems like a good match." Items are scored on a five-point scale of agreement, from 1 = Completely disagree to 5 = Completely agree and averaged for a total score (range: 1–5). An average score of at least 4 will indicate adequate appropriateness with this instrument.

Feasibility relates to the extent to which our I-Score intervention is successfully used or carried out within the study site [59]. To determine feasibility, data will be collected on the consent rate, defined as the proportion of approached eligible patients and physicians who consent to participate. Individuals who choose not to participate will be asked to provide select sociodemographic information (sex, year of birth, preferred language) and their reason(s) with a checklist on a refusal form. If 70% or more agree to participate, the study will be judged feasible on this aspect. We will also examine the retention rate, indicated by the proportion of patients and physicians who complete the study. Eighty percent will be considered the benchmark for success. Missing I-Score data rates due to network failure as well as patient and provider non-completion of self-reported questionnaire data will be calculated as well. The criterion for success on this metric is at least 90% of items completed per participant. Furthermore, participants will complete the Feasibility of Intervention Measure (alpha coefficient: .89) [57], at T1, T2 and T3, a four-item self-report measure that is appropriate for different stakeholder groups (e.g., patients, providers). Example items include "This [evidence-based practice] seems possible" and "This [evidence-based practice] seems doable." Average scores of at least 4 (range: 1–5), indicative of agreement on the 5-point response scale, will signify the I-Score intervention's feasibility.

Fidelity is the degree to which the intervention was implemented as specified in the protocol [59]. It will be indicated by patient and provider adherence to core components of the intervention. Thresholds for success, from T1 to T3, are: 90% patient completion of the I-Score prior to meeting with the physician; 90% provider review of the patient's I-Score results prior to or during the clinic visit.

**Objective 2 -evaluate the implementation strategy.** Evaluation of the implementation strategy will be performed in relation to the same constructs as for the first objective. However, the assessment of acceptability, appropriateness, and fidelity will be solely based on analysis of qualitative data (see Table 2). As to feasibility, it will be assessed in terms of the rate of technical issues encountered and recorded in the Application Manager's notes, and the percentage of providers who participated in the implementation activities (i.e., education meeting, focus groups), with a success threshold set at 80% or more.

**Objective 3 -determine preliminary intervention effectiveness.** This pilot study will collect data on one service outcome, patient management. It will be verified with a checklist submitted to participating physicians to allow them to record, per patient encounter, if they received the I-Score results on time, if they reviewed them prior to or during the clinic visit, if they were discussed during the visit, and if the I-Score identified concerning barriers. Then,

**Table 2. Implementation science metrics and effectiveness outcomes collected for the pilot study.**

| Objective | Construct | Data collected | Threshold for success | Participant group Patients | Participant group Physicians | Timing |
|---|---|---|---|---|---|---|
| Objective 1—Evaluate perceptions of the I-Score innovation | Acceptability | **Primary outcome:** Acceptability E-Scale [54] | Score $M \geq 24$ | ✓ | ✓ | T1, T2, T3 |
| | | % likely to recommend the I-Score [55] | $\geq 80\%$ | ✓ | ✓ | T1, T2, T3 |
| | | Net Promoter Score [55] | $> 0$ | ✓ | ✓ | T1, T2, T3 |
| | Appropriate-ness | Perceived compatibility subscale [56] | Score $M \geq 5.5$ | - | ✓ | T1, T2, T3 |
| | | Appropriateness of Intervention Measure [57] | Score $M \geq 4$ | ✓ | ✓ | T1, T2, T3 |
| | Feasibility | Consent rate (and reasons for refusal) | $\geq 70\%$ | ✓ | ✓ | T1 |
| | | Retention rate | $\geq 80\%$ | ✓ | ✓ | T1, T2, T3 |
| | | Missing PROM (I-Score) data rate (e.g., due to non-completion, network failure) | $\leq 10\%$ | ✓ | ✓ | T1-T3 |
| | | Feasibility of Intervention Measure [57] | Score $M \geq 4$ | ✓ | ✓ | T1, T2, T3 |
| | Fidelity | % patients who complete the I-Score on time | $\geq 90\%$ | ✓ | - | T1, T2, T3 |
| | | % physicians who review the I-Score results on time | $\geq 90\%$ | - | ✓ | T1, T2, T3 |
| Objective 2—Evaluate the implementation strategy | Acceptability | Barriers and facilitators to implementation, based on the qualitative data collected [a] | - | ✓ | ✓ | T1-T3 |
| | Appropriate-ness | Perceived fit of the implementation strategy within the clinic, based on the qualitative data collected [a] | - | ✓ | ✓ | T1-T3 |
| | Feasibility | % of included physicians participating in the implementation activities (educational meeting, focus groups) | $\geq 80\%$ | - | ✓ | T1, T2, T3 |
| | | Rate of technical issues, based on the Application Manager's notes | - | - | - | T1-T3 |
| | Fidelity | How and why the implementation strategy was adapted, based on the qualitative data collected [a] | - | ✓ | ✓ | T1-T3 |
| Objective 3 –Determine preliminary intervention effective-ness | Patient management | Checklist of physician actions following review of the I-Score results | $p \leq 0.05$ | - | ✓ | T1, T2, T3 |
| | Barriers to ART adherence | The I-Score PROM | $p \leq 0.05$ | ✓ | - | T1, T2, T3 |
| | Adherence to ART | Self-Rating Scale Item [58] | $p \leq 0.05$ | ✓ | - | T1, T2, T3 |
| | Viral load | The HIV RNA viral load, as indicated in the patient's medical file ($> 50$ copies/mL = detectable) | $p \leq 0.05$ | ✓ | - | T1, T3 |

PROM: Patient-reported outcome measure; ART: Antiretroviral therapy.

[a] Qualitative data include the Application Manager's notes (T1-T3), the qualitative interviews with patients and the focus groups with physicians (T2, T3).

they will check off any clinical actions that were taken based on the I-Score results or any related patient-provider discussion (e.g., recording issues in medical notes, referring to another health professional, ordering a test, changing a medication or treatment, providing advice or education).

The patient outcomes assessed in this pilot are barriers to adherence, self-reported adherence, and viral load. Barriers to ART adherence will be assessed at T1, T2, and T3, with our

previously described PROM. Adherence will be examined with the Self-Rating Scale Item (SRSI) [58] at T1, T2, and T3. It is a one item measure of treatment adherence (i.e., "Rate your ability to take all your medications as prescribed" [over the past 4 weeks], rated on a 6-point scale (i.e., Very poor, Poor, Fair, Good, Very good, and Excellent). Viral load, a clinical indicator of viral activity (e.g., infectiousness) and treatment response, will be treated as a dichotomous variable based on whether, as indicated in the patient's medical file, the HIV RNA viral load is detectable (over 50 copies), or not. Undetectability is a goal of HIV treatment. The most recent viral load test result at T1 and T3, will be collected.

## Data analysis

The period of qualitative and quantitative data analysis is projected to extend from approximately July 2021 to April 2022.

**Quantitative analysis.** Time 1 questionnaires for people living with HIV and for HIV physicians will be summarized with descriptive statistics. For continuous variables, the minimum, the maximum, the mean, and the standard deviation will be reported. For ordinal and nominal qualitative variables, we will report absolute and relative frequencies (proportions).

As, specified, for most quantitative metrics relating to Objectives 1 and 2, as recommended for pilot studies, score targets were set to evaluate the ability to proceed to a definitive trial [53]. For people living with HIV and HIV physicians, continuous outcomes expressed on a Likert scale will be summarized with the minimum, the maximum, the mean, and the standard deviation at T1, T2 and T3. Binary outcomes (yes or no) will be reported with absolute and relative frequencies (proportions) at T1, T2 and T3. The means and proportions of T1, T2 and T3 will be confronted with their corresponding thresholds for success, presented in Table 2. To study the tendency of means and proportions for people living with HIV over time, a Linear Mixed Model or a Generalized Linear Mixed Model will be used, for continuous and binary outcomes, respectively. The response variable of each model will be the corresponding outcome and the independent variable will be the time (T1, T2 and T3). The null hypothesis of no time effect on the corresponding outcome will be tested with a Student's t-test on the regression coefficient. If the null hypothesis is rejected, we will perform post-hoc Student's paired t-tests between all combinations of time points to show between which time points means and proportions differed significantly. Additionally, to verify if each threshold for success is met at the end of the study, we will test the null hypothesis that each mean or proportion at T3 is inferior to its corresponding threshold with a Student's t-test. For all analyses, a significance level of 5% will be adopted. Finally, where appropriate, Cronbach's alpha will be calculated to evaluate the internal consistency of subscales.

Regarding the patient outcomes of Objective 3, barriers to adherence and adherence will be summarized with the minimum, the maximum, the mean, and the standard deviation. Viral load will be reported by absolute and relative frequencies, as it is considered a dichotomous variable. For the service outcome obtained from the physician checklist, we will report the proportion of clinic visits when physicians took action based on the I-Score results, among the visits where an adherence barrier of concern was identified by physicians. Proportions will be reported for T1, T2 and T3 and globally, across time periods. To evaluate evidence of a statistically significant difference in our chosen effectiveness outcomes, we will run a Student's paired t-test for barriers to adherence and adherence and a McNemar test for viral load, between T1 and T3. To complete the analysis of service outcomes, we will use a logistic regression model, considering only the visits where an adherence barrier of concern was identified by physicians. The dependent variable is the binary variable of whether or not an action was taken by physicians and the independent variable is the time, considered as a factor with three independent

levels (T1, T2 and T3). We will test the null hypothesis that time has no effect on the probability of taking action, with a t-test on the regression coefficient. We will conclude the analysis by testing the null hypothesis of equality of proportions between all pairwise combinations of time points, with a Student's t-test between two proportions, performing a Bonferroni correction for multiple tests. For all analyses, a significance level of 5% will be adopted.

**Qualitative analysis.** The study's qualitative material (i.e., focus groups, interviews, Application Manager notes) will be submitted to content analysis [62], focusing on the manifest content. Deductive content analysis will be favored, allowing implementation barriers and facilitators identified to be categorized with an existing framework, while remaining open to emerging categories. Deductive content analysis allows categories to be compared at different periods, fitting with the study's longitudinal design [63]. For this purpose, the Consolidated Framework for Implementation Research (CFIR) will be used [30]. Analysis will involve three phases [62]: 1) preparation, when the analyst attempts to get a sense of the entire dataset through immersion in the data; 2) organizing, during which an unconstrained categorization matrix will be devised with the CFIR's constructs, and the data will be coded, accordingly. At this point, the qualitative data management software, Atlas.ti version 8, will be used to code and categorize the material; and 3) reporting, which involves presenting the described contents (meanings) of the categories and addressing trustworthiness [62]. A product of these analyses will be matrices of facilitators, barriers and potential solutions raised by patients and physicians, at each main qualitative data collection period, using the CFIR. These will allow for the tracking of categories over time [64], to help identify patterns. Two trained coders will be involved in the qualitative analyses which will be discussed during periodic team meetings, including any discrepancies in coding or interpretation.

For the cyclical small tests of change of the implementation strategy, consistent with the approach by Keith et al. [48], the qualitative data will be coded and categorized with the CFIR. We will further structure and document our cyclical small tests of change by drawing on the iterative Plan-Do-Study-Act (PDSA) approach for quality improvement [65]. During the 'plan' stage, the stakeholder feedback collected will help to periodically identify and document factors that are affecting the intervention and associated changes to the implementation and/ or peripheral components of the intervention that could lead to improvement. Related predictions will be explicitly articulated [65]. During the 'do' stage, changes will be tested. At the 'study' stage, the extent of the change's success will be evaluated against the prediction(s) and documented with subsequent qualitative or quantitative data, per the study's design, and the Application Manager's field notes. The 'act' stage will see further adaptations, depending on the successfulness of the change, and/or the initiation of another cycle of change. For each PDSA cycle undertaken, all decisions and relevant information will be recorded, following the PDSA theoretical framework developed by Taylor et al. [65].

**Mixed methods analysis.** The quantitative and qualitative data will be analyzed separately and subsequently brought together for comparison, for a more complete interpretation of the results. Areas of convergence and divergence will be highlighted.

## Data management

The Opal team (TH, YM) will oversee data management for this study. Patient-reported data will be electronically collected directly through Opal. This data will be stored in a local server protected by the MUHC. The Opal team will manage patient information through the Opal app, always respecting data security and confidentiality. For this study's purpose, relevant dei-dentified data will be extracted by the team and stored on a password protected USB key or hard drive for subsequent analysis.

## Discussion

Electronic PROM use for clinical practice with individuals living with HIV is limited. To our knowledge, this is the first pilot study of an intervention aimed at implementing systematic consideration of patient-identified ART adherence barriers with an electronic PROM in routine HIV care. This pilot study will help build needed knowledge on impediments to and strategies for implementing these tools in HIV care [1]. As such, it acts as a standalone study, providing useful and rich data to others considering similar interventions in similar contexts. It will generate data that will improve understanding of conditions for successful implementation as well as test and solidify the implementation strategy. Furthermore, it may shed light on the mechanisms of similar PROM interventions. Overall, it will produce useful data to design a definitive effectiveness trial of the I-Score intervention.

### Anticipated problems

There are many potential barriers to implementing PROMs in care, such as provider reticence (e.g., due to concerns for increased workload). Our multi-pronged implementation strategy directly seeks to mitigate numerous common barriers to implementing PROMs in clinical practice [43]. For details, see S1 Table. Also in our favor is the study site (CVIS), which is a highly active center for HIV research where several of its investigators are experienced in implementation science methods.

   With the use of self-reported adherence measures, there are accuracy concerns. These measures are known for being prone to recall bias, if not misremembering, given, for instance, the mundane nature of medication-taking [66,67]. They are also deemed vulnerable to social desirability bias, if not intentional deception, given, for example, patient beliefs about the consequences of admitting adherence problems [66,67]. These processes could influence the types of adherence barriers people are able or willing to report when completing our PROM. Conversely, PROM administration in routine care is recognized to give patients permission to raise health problems with their providers [68]. Furthermore, our understanding is that barriers are multidimensional and interconnected, where a common barrier such as forgetting, can be associated with numerous others (e.g., substance use, HIV stigma, life demands, co-morbidity) [20]. Hence, while our PROM is not expected to capture the full details of an individual's barriers, a fuller portrait may emerge through conversations the PROM instigates with providers. Our team has also vied to involve people living with HIV with a range of methods throughout the development of our instrument (via committee meetings, cognitive interviewing, etc.) to ensure its relevance, acceptability, and appropriateness for use in HIV care. This pilot study will allow us to further gauge the utility of the information provided by the PROM and to potentially make adjustments to improve accuracy.

   Finally, an added concern is the continued spread of COVID-19. Physicians have been advised to use telemedicine and teleconsultations, whenever possible, to limit the spread of COVID-19 [69,70]. Given the uncertain evolution of the pandemic and associated public health response, methodological adjustments to this study may be required, for instance, to further limit in-person participant visits with physicians and research team members.

## Conclusion

The PROM initiative concerned by this study challenges traditional care paradigms with a more patient-centered approach. It aims to shift an HIV treatment paradigm emphasizing biomedical markers (i.e., viral load) in adherence management. Systematic monitoring of patient-reported adherence barriers could allow for a more preventative approach and help ensure adherence management addresses patients' priorities. Indeed, the I-Score PROM includes

only the most highly valued barriers in terms of relevance and importance to HIV care, as rated by people living with HIV and providers in our Delphi consultation [42]. As to the app through which the PROM is administered and its features, it may help redress the patient-provider knowledge imbalance and empower patients in their care [71].

## Supporting information

**S1 Checklist. SPIRIT 2013 checklist: Recommended items to address in a clinical trial protocol and related documents**[*]**.**
(DOC)

**S1 Table. Relationship between the identified facilitators/barriers of PROM implementation, the implementation framework (CFIR), and the pilot study's implementation strategies.** PROM = patient-reported outcome measure; CFIR = Consolidated Framework for Implementation Research; [a] Reproduced or adapted from Foster et al. [41]; [b] Based on Damschroder et al. [25]; [c] Based on the taxonomies of Powell et al. [42,43].
(DOCX)

**S1 Appendix. Patient consent form.**
(DOCX)

**S2 Appendix. Patient and physician Time 1 study questionnaires.**
(DOCX)

**S1 Dataset. World Health Organization Trial Registration Data Set for study CTNPT039.**
(DOCX)

**S1 File. The I-Score/Opal implementation pilot study.**
(PDF)

## Acknowledgments

We wish to thank the patients and health and social service providers who participated in the development of the I-Score PROM and in adapting Opal for HIV care. On these projects, we also thank the Quebec SPOR Support Unit -McGill Methodological Developments Platform, for sharing their expertise and resources.

## Author Contributions

**Conceptualization:** Kim Engler, Serge Vicente, Bertrand Lebouché.

**Funding acquisition:** Kim Engler, Sara Ahmed, Marina Klein, Bertrand Lebouché.

**Methodology:** Kim Engler, Serge Vicente, Yuanchao Ma, Joseph Cox, Sara Ahmed, Marina Klein, Bertrand Lebouché.

**Software:** Yuanchao Ma, Tarek Hijal, Sofiane Achiche.

**Writing – original draft:** Kim Engler, Serge Vicente.

**Writing – review & editing:** Kim Engler, Serge Vicente, Yuanchao Ma, Tarek Hijal, Joseph Cox, Sara Ahmed, Marina Klein, Sofiane Achiche, Nitika Pant Pai, Alexandra de Pokomandy, Karine Lacombe, Bertrand Lebouché.

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
