## [Decision Letter · Decision Letter 0]

25 May 2021

PONE-D-20-37626

Implementation of an electronic patient-reported measure of barriers to antiretroviral therapy adherence with the Opal patient portal: a mixed method type 3 hybrid pilot study at a large Montreal HIV clinic

PLOS ONE

Dear Dr. Engler,

Thank you for submitting your manuscript to PLOS ONE. After careful consideration, we feel that it has merit but does not fully meet PLOS ONE’s publication criteria as it currently stands. Therefore, we invite you to submit a revised version of the manuscript that addresses the points raised during the review process. As you will see, two reviewers were impressed with your study and recognized its potential impact on the field. Our third reviewer had some concerns about moving forward with this publication prior to obtaining more data. I do not think we should postpone this work, however they do raise important concerns that are worth addressing. Please attend to all of their comments as I think these will be readily addressable and should result in an improved manuscript.

We look forward to receiving your revised manuscript.

Kind regards,

Ethan Moitra

Academic Editor

PLOS ONE

Brown University

Journal Requirements:

"BL, KE, SA, and MK received pilot funding from the CIHR Canadian HIV Trials Network

(https://www.hivnet.ubc.ca/) to conduct this study (Grant # CTNPT039).

BL, KE received funding from Merck Canada Inc. to conduct the I-Score PROM development study under their Investigator-Initiated Study Program (Grant # IISP-53538). The funders had and will not have a decisional role in study design, data collection, data analysis, or preparation of manuscripts for publication based on its results. "

We note that you received funding from a commercial source: Merck Canada Inc.

Reviewers' comments:

Reviewer's Responses to Questions

**Comments to the Author**

1. Does the manuscript provide a valid rationale for the proposed study, with clearly identified and justified research questions?

Reviewer #1: Yes

Reviewer #2: Yes

2. Is the protocol technically sound and planned in a manner that will lead to a meaningful outcome and allow testing the stated hypotheses?

Reviewer #1: Yes

Reviewer #2: Yes

3. Is the methodology feasible and described in sufficient detail to allow the work to be replicable?

Reviewer #1: Yes

Reviewer #2: Yes

4. Have the authors described where all data underlying the findings will be made available when the study is complete?

Reviewer #1: Yes

Reviewer #2: Yes

5. Is the manuscript presented in an intelligible fashion and written in standard English?

Reviewer #1: Yes

Reviewer #2: Yes

6. Review Comments to the Author

You may also provide optional suggestions and comments to authors that they might find helpful in planning their study.

Reviewer #1: This was a really interesting and detailed protocol of a pilot hybrid mixed methods study. I look forward to reading the results of the pilot and potentially the final trial when it is ready. It may be good to use the term protocol in your title since it isn't immediately clear. In your article, you showed knowledge of implementation research and mixed methods design and analysis of data. The pilot is complicated so even though you included sufficient amounts of detail, I did have to keep going back and forth between sections to clarify any questions I had.

Overall, I think this is a good sound study. I do have a few minor questions /comments below:

1) It would be good to keep consistent with the use of acronyms. Sometimes you use PROM and sometimes you use "patient-reported outcome measures" (e.g. on line 87 and on line 97).

2) Line 134 - You mention how many patients are in the HIV clinic, but it would also be good to mention how many physicians are in the clinic. It is interesting to know since you mention your sample size for the pilot will be 5 physicians.

3) Line 139 - Is there a inclusion criteria related to length of time on ART? You mention later that participants have to have signs of adherence problems within the last 12 months, but it wasn't clear if they had to have been taking ART for that long.

4) Line 156 - Many studies have physicians involved in recruiting their patients, but I just wondered if you had considered the possible ethical issues associated with this? Patients may feel undue pressure or obligation to participate.

5) Line 188 - Table S1 was not available or I was not able to find it so I was unable to review

6) Line 193 - Will the Application Manager be the one meeting with the patients/providers before T1 to train them?

7) Line 207 - With the cyclical tests of change, does this mean the process may be adapted for participants e.g. from T2 to T3? Would the change occur for all participants or be on an individual basis?

8) More detail on the researchers involved would be useful and interesting. Who will be running the focus groups? Who and how many researchers will be analysing the data? How will disagreements between researchers be handled? Particularly for qualitative data, having an overview of the background of the researchers is important.

9) Table 1 (Line 238) - At the very end it mentions compensation for the patients, but I was unable to find any detail on what this compensation was.

10) When reviewing Table 2 (Line 248), I was wondering how long approximately will the questionnaires take at the different times points for both patients and physicians? It was also not exactly clear whether the questionnaires were filled in before the patient meets their physician or afterwards at each time point.

11) Line 330 - change qualitative to quantitative

12) You do mention that you have published on the barriers measure elsewhere, but it would have been good to have included a little more detail on this - how many barriers and the type of barriers it covers.

Well done again on a great paper! Good luck with the study!

Reviewer #2: The protocol is clearly written and robust and only minor comments are attached.

Minor Comments:

1. It is a awkward to write (ln. 49) that PROMs are rarely used in clinical practice and then state (ln. 54) that evidence for PROM is mixed and then beneficial in the following paragraph. Unclear if referring to PROMs overall or PROMs related to HIV.

2. The team is investigating barriers to ART adherence and cite some recently literature on the measurement of this topic. However, as others have shown, one major limitation is relying on participants to describe what are their most important barriers, which may be based on convenience and social desirability. For example, studies show that the #1 barrier to ART adherence is "simply forgetting," irrespective of viral load, thus, it could be that simply forgetting is a consequence of neurocognitive functioning, depression, or social desirability in not wanting to report substance use. The promise of an PROM intervention must take into account why PLWH report the barriers they do as you may miss the targets that truly drive non-adherence.

7. PLOS authors have the option to publish the peer review history of their article (what does this mean?). If published, this will include your full peer review and any attached files.

Reviewer #1: No

Reviewer #2: No

---

## [Author Response · Author response to Decision Letter 0]

8 Jun 2021

My response to the reviewers can be found in the documents attached.

---

## [Decision Letter · Decision Letter 1]

23 Nov 2021

Implementation of an electronic patient-reported measure of barriers to antiretroviral therapy adherence with the Opal patient portal: protocol for a mixed method type 3 hybrid pilot study at a large Montreal HIV clinic

PONE-D-20-37626R1

Dear Dr. Engler,

We’re pleased to inform you that your manuscript has been judged scientifically suitable for publication and will be formally accepted for publication once it meets all outstanding technical requirements. A reviewer confirmed that you addressed all of their comments. Unfortunately, one of the other reviewers was not available for a second review. As such, I assessed your responsiveness to this person's comments and I think they were addressed. Lastly, our statistical reviewer (reviewer #3) continued to raise concerns about proceeding to publication now vs. when you enroll participants. I think these are helpful comments and I would encourage you to consider them as you move forward in this study. However, given your focus on the protocol of this study, rather than the findings, I believe this manuscript is suitable for publication.

Kind regards,

Ethan Moitra

Academic Editor

PLOS ONE

Brown University

Reviewers' comments:

Reviewer's Responses to Questions

**Comments to the Author**

1. Does the manuscript provide a valid rationale for the proposed study, with clearly identified and justified research questions?

Reviewer #2: Yes

Reviewer #3: Yes

2. Is the protocol technically sound and planned in a manner that will lead to a meaningful outcome and allow testing the stated hypotheses?

Reviewer #2: Yes

Reviewer #3: Partly

3. Is the methodology feasible and described in sufficient detail to allow the work to be replicable?

Reviewer #2: Yes

Reviewer #3: Yes

4. Have the authors described where all data underlying the findings will be made available when the study is complete?

Reviewer #2: Yes

Reviewer #3: No

5. Is the manuscript presented in an intelligible fashion and written in standard English?

Reviewer #2: Yes

Reviewer #3: Yes

6. Review Comments to the Author

You may also provide optional suggestions and comments to authors that they might find helpful in planning their study.

Reviewer #2: The authors have adequately responded to all comments raised by the reviewers. I have no further comments.

Reviewer #3: This is a very well written report of describing the protocol of a pilot study aiming to evaluate patients’ and physicians’ perceptions of the I-Score intervention and its implementation strategy.

From reading the text there is a sense that the COVID-19 is impacting on the feasibility of this very well designed project even of this pilot phase with a small sample size of n=30. Data were supposed to be collected by end of February then extended to April-August 2021. It is now mid May and no data are yet available. Although the publication of the protocol might be useful in its own right, I suggest submission is postponed until the firsts 30 patients are included and some data are shown.

Some comments to the implementation plan below

1. Unclear why the plan is to recruit only PLWH with known or suspected adherence problems. It might be useful to have a control group of PLWH with current VL≤50 to see whether they also might show fatigue between T1 and T3 and to compare patterns with those of the suspected non-adherent.

2. I think that there should be a plan for how to control for possible collider bias. Included population will be a selected sample of PLWH who own a smartphone with an appropriate data plan and/or home Wi-Fi connection as the Opal app is suited to a smartphone interface. People with smartphone are typically different from those who do not own one, being younger more literate and with higher self-health awareness. These factors can also be causes of the outcomes of interests thus introducing collider bias. There should be a plan for extracting a representative sample of the universe population at the outset to be able to compare characteristics of included and excluded and in case perform weighted analysis to control for collider bias.

3. Unclear why paper questionnaires are needed at all. Could not everything be collected through the app?

4. Unclear how Zoom meetings will be kept anonymised and who will be able to access the recordings and through which platform.

5. Because the pilot study only has a sample size of 30, the threshold of p=0.05 for objective 3 seems to be too high. Assume 0.05 if the type I error for the final analysis and this could be seen as an interim analysis after 30 people are in so I would correct for type I error inflation

6. I wonder whether adherence should be collected through more validated tools such as the VAS scale (percentage of missed dose over the previous 4 weeks). Not sure whether it is implementable in the Opal app but should not be too complicated (a bar going from 0-100% in which a position could be pointed at?). This would guarantee to collect a more accurate value than a 5-scale ordered category, potentially reducing misclassification of exposure and residual confounding

7. Objective 3 seems to be structured such as every participant is a control of self. Because of the relatively short follow-up this might lead to low power of paired tests (most people will probably show flat trajectories at least in the quantitative measures). It might be useful to have a control group to whom the app was not available at all to compare results across groups and better evaluate the effectiveness of the intervention. Although randomisation would be ideal, it could be done as an observational study by controlling for key confounders.

Minor points

Lines 89 and 286: is ‘select’ a word? Should it be ‘selected’ instead?

7. PLOS authors have the option to publish the peer review history of their article (what does this mean?). If published, this will include your full peer review and any attached files.

Reviewer #2: No

Reviewer #3: No

---

## [Editor Report · Acceptance letter]

20 Dec 2021

PONE-D-20-37626R1 

Implementation of an electronic patient-reported measure of barriers to antiretroviral therapy adherence with the Opal patient portal: protocol for a mixed method type 3 hybrid pilot study at a large Montreal HIV clinic 

Dear Dr. Engler:

I'm pleased to inform you that your manuscript has been deemed suitable for publication in PLOS ONE. Congratulations! Your manuscript is now with our production department. 

Kind regards, 

on behalf of

Dr. Ethan Moitra 

Academic Editor

PLOS ONE